# Glioblastoma’s Next Top Model: Novel Culture Systems for Brain Cancer Radiotherapy Research

**DOI:** 10.3390/cancers11010044

**Published:** 2019-01-04

**Authors:** Seamus Caragher, Anthony J. Chalmers, Natividad Gomez-Roman

**Affiliations:** Institute of Cancer Sciences, University of Glasgow, Glasgow G61 1QH, UK; seamuscaragher@gmail.com (S.C.); anthony.chalmers@glasgow.ac.uk (A.J.C.)

**Keywords:** glioma, glioblastoma, pre-clinical drug development, 3D culture systems, radiotherapy

## Abstract

Glioblastoma (GBM), the most common and aggressive primary brain tumor in adults, remains one of the least treatable cancers. Current standard of care—combining surgical resection, radiation, and alkylating chemotherapy—results in a median survival of only 15 months. Despite decades of investment and research into the development of new therapies, most candidate anti-glioma compounds fail to translate into effective treatments in clinical trials. One key issue underlying this failure of therapies that work in pre-clinical models to generate meaningful improvement in human patients is the profound mismatch between drug discovery systems—cell cultures and mouse models—and the actual tumors they are supposed to imitate. Indeed, current strategies that evaluate the effects of novel treatments on GBM cells in vitro fail to account for a wide range of factors known to influence tumor growth. These include secreted factors, the brain’s unique extracellular matrix, circulatory structures, the presence of non-tumor brain cells, and nutrient sources available for tumor metabolism. While mouse models provide a more realistic testing ground for potential therapies, they still fail to account for the full complexity of tumor-microenvironment interactions, as well as the role of the immune system. Based on the limitations of current models, researchers have begun to develop and implement novel culture systems that better recapitulate the complex reality of brain tumors growing in situ. A rise in the use of patient derived cells, creative combinations of added growth factors and supplements, may provide a more effective proving ground for the development of novel therapies. This review will summarize and analyze these exciting developments in 3D culturing systems. Special attention will be paid to how they enhance the design and identification of compounds that increase the efficacy of radiotherapy, a bedrock of GBM treatment.

## 1. Introduction

In glioblastoma (GBM), we have seen the disastrous result of insufficiently realistic drug screening and testing systems. From cilengitide to target therapies, drugs developed using contemporary pre-clinical systems have roundly failed to predict clinical efficacy. We believe that a large part of the blame for these failures should be laid at the feet of our overly simplified in vitro culture and murine systems. At every level, the defining characteristic of GBM is complexity. From genetic anomalies and epigenetic dysregulation [1,2] to cellular plasticity [3], immunosuppression [4], and metabolic adaptability [5,6,7], GBM exhibit a wide diversity of phenotypes and possess a remarkable toolbox for colonizing the brain and resisting current treatment regimens. The failure of current radiotherapy paradigms to provide long-term tumor control in GBM has been linked to all these factors [8]. In addition to these tumor factors, GBM remains intractable because of its ability to interact with its environment, including healthy astrocytes and blood vessels. The blood-brain-barrier further complicates therapeutic design. The challenge of pre-clinical drug development is to create culture systems and animal models that faithfully recapitulate these complex behaviors and systems, thereby enabling researchers to identify compounds that effectively translate success targeting cancer cells in vitro models to meaningful improvement for patients. This review will analyze the limitations of current GBM drug development models and summarize and discuss novel culture systems under development.

## 2. Limits of Current Pre-Clinical Systems

In order to discuss the potential of novel culture systems to more effectively ascertain the clinical potential of candidate compounds, we must first discuss the current state of GBM culture and murine models. Further, we must delineate how the immense complexity of brain tumor genetics and the vast influence of the brain microenvironment limit their utility. Figure 1 illustrates the current mismatch between tumors growing in patients and present culture protocols.

### 2.1. Current Practices for Pre-Clinical GBM Modeling

At present, the gold standard of GBM in vitro culture systems relies on the use of patient derived (PD) cells. Human immortalized glioma cells, such as U87 and U251, have fallen out of favor due to lack of similarity to actual GBM tumor cells and controversy regarding their origins [9]. In their place, a variety of PD lines have been developed. These cells are derived from the tumor tissue removed during surgical resection, which is processed to generate cell lines that can be passaged through the flanks of immunodefficient mice and cultured as monolayers or in suspension. A wide number of these cell lines exist, usually created at academic institutions and then shared through collaborations across the neuro-oncology research network.

A range of culture conditions are used to propagate these cells once they are growing in vitro. The different choices of media and additives were recently reviewed [10]. A survey of the literature demonstrated that the majority of labs currently employ Dulbecco’s Modified Eagle Medium (DMEM) on its own or in combination with F12, a nutrient mixture (SigmaAldrich, St Louis, MO, USA). This media contains a mix of glucose, amino acids, inorganic salts, and vitamins. In addition, many groups now add growth factors to their media, most commonly epidermial growth factor (EGF) and/or basic fibroblast growth factor (bFGF). The addition of these growth factors has been shown to promote maintenance of gene expression characteristics observed in human patient samples and proliferation [11]. Finally, the use of supplements like N2 and B27 is continuing to rise in GBM culture. The main component of N2 is putrescine, a diamine generated in the decomposition of amino acids (ThermoFisher, Waltham, MA, USA). The exact composition of B27 is proprietary, but it contains a variety of lipid compounds, including linoleic acid, corticosterone, and progesterone (ThermoFisher). As we will see, the composition of media can have a profound effect on the phenotype of GBM cells, making culture conditions a key factor limiting the efficacy of current models.

Cells grown as monolayers undergo profound phenotypical changes and have been reported to exhibit markedly different responses to cytotoxic treatments than those observed in patients [12,13,14]. In the context of radiation therapy, 3D culture of lung and head & neck cancer cells embedded in laminin-rich extracellular matrix (ECM) has been shown to promote radiation resistance compared to 2D culture by chromatin condensation, facilitating repair of DNA double strand breaks (DSB) [12,15,16]. Colorectal cancer cell lines cultured in similar laminin-rich ECM 3D conditions also exhibited changes in cellular morphology, phenotype and gene expression and were resistant to treatment with epidermal growth factor receptor (EGFR) inhibitors compared to cells cultured in 2D conditions [13,17]. In the context of GBM, cells exhibited differential radiosensitisation to EGFR and VEGF inhibitors in 2D and 3D systems [18]. These observations provide a potential explanation for the frequent failure of results derived from conventional 2D cell culture systems to predict clinical efficacy [19,20], a phenomenon that leads to inappropriate and excessive use of animal testing.

### 2.2. Tumor Intrinsic Factors

Each GBM tumor is highly unique in terms of genetic etiology and gene expression profile. In addition, we know from decades of research that GBM is a highly dynamic tumor type, rapidly adapting to changes in its surroundings. As such, any effort to model this cancer in the dish must rely on careful consideration of a number of variables when initiating cultures and selecting culture conditions.

#### 2.2.1. Inter and Intratumoral Genetic Heterogeneity

Genetic variation is a hallmark of GBM. A landmark paper, examining over 400 primary patient samples, identified four subtypes of tumors based on their genomic characteristics. Many of the mutations that defined the groups occurred in receptor tyrosine kinases, including epidermal growth factor receptor (EGFR) and platelet derived growth factor receptor (PDGFR) [2]. Subsequent research has reduced the four subtypes of tumor to three groups—classical, proneural, and mesenchymal—although the clinical relevance of these classifications remains uncertain. Further complicating the matter, recent reports utilizing advances in single cell RNA sequencing have demonstrated that many patients’ tumors actually contain cells of all three subtypes [21]. This intratumoral heterogeneity includes the key growth factors, with reports of mosaic amplification of several receptor tyrosine kinases [22], as well as subpopulations within tumors expressing a multitude of variants for a single kinase [1]. This diversity also involves the presence of a subpopulation of tumor cells termed glioma initiating cells or glioma stem-like cells (GSCs). GSCs take their name from the fact that they display many of the key features of healthy neural stem cells, including the ability to self-renew, the capacity to generate progeny of a variety of different cell types, and expression of key stem cell related genes [23,24]

This genetic heterogeneity complicates the process of modeling GBM in vitro. First, each patient-derived (PD) cell line represents only one of the multitudes of genetic permutations that can underlie a GBM tumor. For example, a panel of PD GBM lines originally developed by a group at the Mayo Clinic varied greatly across each sample in terms of well-established GBM mutations, including epidermal growth factor receptor (EGFR), phosphatase and tensin homologue (PTEN), p53, and platelet derived growth factor receptors A (PDGFRA) [25]. While the status of these mutations in each tumor were maintained in murine intracranial xenografts, the fact remains that selecting a PD cell line inherently for experiments fundamentally biases the experiment. A similar set of cell lines, with a range of genetic and phenotypic profiles, has been generated by a group at the University of Cambridge [26]. In light of this bias, it is critical that a variety of cell lines be utilized for all experiments, in order to guard against biased results. These panels will also provide critical insight into potential predictive biomarkers, as efficacy in cell lines only bearing certain specific mutations would point towards potential use in select patient populations. Second, the process of deriving a cell line from patient tumors is inherently a selection process. Since the first attempts to culture brain tumor cells from surgical biopsies in 1929, only certain samples have been capable of growing in vitro [27]. The specific clones of each tumor that survive and grow in culture will automatically skew the results of assays, as these clones represent only a certain subpopulation of the tumor. In order to compensate for this issue of intertumoral heterogeneity, labs must perform all their pre-clinical assays and animal modeling using multiple cell lines of different subtypes. This factor inherently multiples the resources and time required to explore new therapies for GBM.

In terms of intratumoral diversity, the process of culturing GBM cells remains incredibly complicated. Evidence increasingly shows that different subpopulations of tumor cells have a unique role to play in tumor maintenance and progression. For example, it has recently been shown that differentiated GBM cells communicate bi-directionally with GSCs, promoting GBM progression [28]. The selection process inherent in deriving cells from patient samples carries an extremely high risk of losing certain tumor subpopulations, and subsequently the loss of subpopulation interactions critical for tumor behavior. Current culture techniques often promote homogeneity in cells, thereby limiting their utility. As we shall discuss, some novel in vitro techniques have improved capacities for the maintenance of a heterogeneous cell mixture.

#### 2.2.2. Cellular Plasticity

Further complicating the generation of representative tumor cultures, GBM cells have demonstrated a profound ability to adapt their phenotypes to their surroundings. Rather than existing as a stable population, GBM cells exist in a state of dynamic equilibrium [3]. Termed cellular plasticity, this process means that brain tumors are a moving target, capable of acquiring multiple different phenotypes. Microenvironmental cues from acidity and hypoxia, as well as the stress caused by chemotherapy and radiation, can alter the gene expression and functional state of GBM cells, including induction of the GSC state [3,29,30,31,32,33,34,35]. Research continues to identify novel environmental activators of cellular plasticity. A recent report highlighted extracellular vesicles secreted from apoptotic GBM cells transfer splicing factors to surviving neighbors, altering recipient cell transcription and shifting them to a more aggressive phenotype [36].

The sensitivity of GBM cells to their surroundings has implications for the process of generating useful in vitro models of brain tumors, because even subtle changes in culture conditions may have a high degree of influence on phenotype. For example, the addition of fetal bovine serum (FBS), a traditional additive to media for cell cultures, has been shown to drastically alter the phenotype of patient derived GBM cells. Specifically, the presence of serum led to the loss of GSC subpopulation and the outgrowth of cells with novel genetic and functional characteristic that did not match the initially isolated sample [11]. This study provided the impetus for the adoption of the serum-free, growth factor containing cultures in use today. However, it has since been shown that slight changes in the concentration of various additive growth factors can alter cell behavior and drug sensitivity (recently reviewed [10]). In addition, the act of passaging cells itself has been shown to activate cellular plasticity, altering the gene expression and phenotype of patient derived GBM cells, including drug sensitivity [37]. For cell lines that are passaged through mouse flanks or brain, it has been shown that these processes cause mouse specific changes in tumor phenotype and gene expression, again leading to inconsistent sensitivity to drugs [38]. Clearly, the adaptability of GBM cells to their environment means that great care must be taken to maximize consistency within labs and across research groups.

### 2.3. Microenvironmental Factors

The brain represents one of the most complicated organ systems in the human body. Its function relies heavily on the cultivation of a microenvironment that promotes effective communication between and among astrocytes and neurons. As a result, the brain utilizes an outsized amount of glucose to enable the energetically expensive processes of cognition [39]. In addition, the cellular microenvironment is more regulated than in the periphery, with blood vessels and transport of nutrients into the brain being tightly controlled by astrocytes and other cellular and non-cellular components [40]. Termed the blood brain barrier (BBB), these structures enable the creation of a highly specific molecular milieu within which GBM must survive and proliferate, and in turn complicates GBM therapy (recently reviewed [41]). From nutrients to growth factors, the brain is a unique place for cancer to occupy (expertly reviewed [42]). Many of these factors, however, are not currently accounted for in culture systems.

#### 2.3.1. Brain Extracellular Matrix

One of the unique features of the brain is the composition of the extracellular matrix (ECM). While the ECM of many tissues contains large amounts of collagen and laminin, the brain’s ECM is relatively low in these fibrous ECM component, mainly found surrounding blood vessels, and more defined by the presence of large amounts of specific proteoglycans and glycoproteins [43]. It has been known for decades that the brain contains elevated levels of hyaluronic acid and heparin sulfate proteoglycans [44]. Thus, culture coating techniques borrowed from research on other organs or tumor types will likely fail to recapitulate the composition of the brain microenvironment. Experimental evidence confirms that differences in the ECM are anything but trivial. GBM cells are sensitive to the stiffness of their surroundings, responding to different environments with changes in invasive capacity and proliferation rates [45]. An analysis of patient samples confirms that the stiffness plays a key role in GBM, with ECM stiffness positively correlated with tumor aggressiveness [46]. Thus, the selection of plastic and coatings for cultures represent a key question in designing cultures for GBM. Further complicating the process of modeling GBM, the ECM surrounding tumors varies widely depending on the portion of the tumor examined and influences progression [47]. Current culture techniques fail to account for either the unique ECM of the brain or its heterogeneity across tumor compartments. Coated flask will have the same coating over the full surface area, thereby creating a homogenous environment. In light of the connection between microenvironmental factors and cellular phenotype, these homogeneous surroundings may affect the behavior of GBM cells and push them towards an artificial state in vitro, limiting the utility of current models for pre-clinical drug development.

#### 2.3.2. Regional Differences in the Tumor Microenvironment

These differences across tumor compartments extend beyond the composition of the ECM. Several different anatomic areas, characterized by unique environmental components and the presence of distinct subpopulations of tumor cells, have been described, including the necrotic core, perinecrotic zone, cellular tumor, leading edge, infiltrating tumor, pseudopalisading cells, hyperplastic blood vessels, and microvessel proliferating cells. Indeed, in situ hybridization studies from the Allen Institute illustrated the intense heterogeneity of gene expression across individual tumors [48].These areas are known to be formed of, and indeed to promote the formation of, specific subpopulations of cells. Critically, GSC preferentially occupy the perivascular niche and the perinectrotic zone [49,50].

Vascularization of tumors is another key factor in microenvironmental heterogeneity. GBM exhibit frequent and severe hypoxia and secrete high levels of pro-angiongenic factors, leading to the formation of a large network of irregular blood vessles [51,52]. These factors, combined with the explosive growth rates of many GBM, result in variable oxygen availability throughout the tumor [53,54]. Given that hypoxia has been well established to alter the gene expression and phenotype of GBM cells, these regional differences contribute to the heterogeneous nature of brain tumors [29,35,55,56].

Current cell culture techniques fail to account for these regional differences in the surroundings encountered by GBM tumors. As most cultures are monolayers in petri dishes or culture flasks, the availability of oxygen and nutrients is uniform based on simple diffusion. While hypoxic chambers have been used to examine the effects of hypoxia on GBM cells [34,35,55], these experiments alter the conditions of the *entire* population, rather than recapitulating the wide range of oxygen availabilities across the tumor microenvironment.

#### 2.3.3. Nutrient Availability and Metabolism

Since the identification of the Warburg Effect, the metabolic behavior of tumors has been of intense interest to cancer researchers. Specifically in the case of GBM, this topic has gained more attention in recent years because of its connection to tumor heterogeneity. Metabolomic and proteomic analysis of 17 PD glioma stem cells indicated that, as with genetic anomalies, GBM exhibit a heterogeneous range of metabolic phenotypes, which appear to cluster into two groups—a proneural-like group characterized by neurotransmitter metabolites and a mesenchymal-like group, characterized by elevated lipid metabolism [7]. In addition, metabolic behavior in GBM has been linked to tumor growth, cell signaling, and epigenetic regulation of gene expression [57,58,59,60]. The role of metabolism in GBM further complicates efforts to faithfully model this tumor in vitro. As mentioned previously, the majority of GBM culturing occurs in DMEM media, which contains approximately 25 mM glucose (manufacturer’s website). In contrast, normal brain extracellular glucose levels are approximately 4.5 mM [61]. It is understandable that media contains higher levels of glucose in order to ensure sufficient energy supply in the absence of blood supply. However, these initial high levels may alter the state of cells and influence cellular response in assays and drug screens. Indeed, a recent report showed that glucose concentration alters the activity of many key kinases in GBM cells; sensitivity to inhibitors with therapeutic potential varied widely based on glucose availability [62]. The importance of understanding the dynamic nature of the tumor microenvironment is discussed in detail below. In addition, current nutrient supplements (F12, B27, and N2) provide only a selection of lipid and fatty acid compounds. Many studies have shown that nutrient availability and use exert great influence over GBM cells [39,63]. This means that media composition, with elevated glucose and limited diversity of other energy sources, is likely to alter GBM phenotype and generate in vitro models that do not reflect actual tumors.

#### 2.3.4. Non-Tumor Cells: Secreted Factors and Contact-Mediated Interactions

Another key factor influencing the behavior of GBM is the presence of neighboring non-tumor cells, which can modulate tumor cells and the microenvironmental milieu via both the secretion of factors and direct contact. These cell types include astrocytes, neurons, endothelial cells, and brain-resident microglia, as well as peripheral immune cells that have entered tumors (though the exact extent of peripheral immune involvement remains a topic of much debate).

Astrocytes, the most abundant cells in the human brain, play a key role in supporting neurons and sculpting the microenvironment under normal conditions (reviewed [64]). Astrocyte co-culture experiments reveal that these same processes may participate in the onset and progression of glioblastoma. For example, a recent report demonstrated that GBM tumor cells are highly dependent on cholesterol secreted by nearby astrocytes for their survival [65]. Astrocytes have also been shown to promote invasion by GSCs via the secretion of cytokines [66]. Co-culture experiments have provided more insight into the role of healthy astrocytes in influencing GBM progression. GBM cells grown in mixed culture with astrocytes were less sensitive to radiation via chemokine release and activation of STAT3 [67]. Reactive gliosis, a process by which astrocytes become activated following brain injury and begin to secrete a variety of factors, including chemokines, is well-established in the peri-tumoral area [68]. It has been demonstrated that this reactive gliosis increases tumor proliferation and invasion in mouse models [69]. In addition to secreted factors, a number of studies have shown that astrocytes physically interconnect with brain metastases and promote tumor colonization [70]. We suspect it is only a matter of time before similar evidence emerges linking GBM growth to contact with neighboring cells. In light of the lack of astrocytes in current in vitro culture protocols, it is unsurprising that they have roundly failed to predict the efficacy of new radiosensitizer drugs in human patients, in whom astrocytes are likely exert radio-protective effects.

Neurons interact with one another through the secretion of neurotransmitters and other factors from the synaptic cleft. Though the bulk of vesicular release occurs in the so-called “active site” of the synaptic cleft, a certain amount of secreted factors spill over from the cleft [71]. Neurons and astrocytes express a variety of receptors and pumps to deal with these extra-synaptic molecules and rely on them for regulation of neuronal activity and astrocyte behavior [72,73]. Evidence is beginning to accrue demonstrating that these extra-synaptic molecules can influence GBM progression. For example, it was recently shown that neuroligin 3, secreted from tumor-promixate neurons, augments tumor growth and creates a positive feedback loop leading to the generation of tumor-derived neuroligin-3 [74,75]. In addition, tumor cells are known to secrete glutamate in order to further activate neuronal activity, which can in turn promote their growth [76,77]. A variety of other neurotransmitters have been suggested as potential players in GBM development, including the monoamines [78]. Again, we suspect that the lack of neurons and the molecules they contribute to the tumor microenvironment helps explain the lack of connection between in vitro drug screens and clinical effect.

Endothelial cells from tumor-associated blood vessels also exhibit bidirectional communication with tumor cells. As previously mentioned, GSC preferentially occupy the perivascular niche, which contains factors that promote their stem-like phenotype and may also enhance resistance [50]. Furthermore, endothelial cells are known to express membrane bound proteins that enable GBM cells to invade the surrounding brain parenchyma along blood vessels [79]. Conversely, GBM cells express high levels of vascular endothelial growth factor (VEGF), which promotes the formation of new blood vessels [80,81,82]. Clearly, endothelial cells exercise influence on tumors via secreted factors and contact, as well as through their influence on the availability of oxygen and nutrients. The lack of endothelial cells in current in vitro models of GBM therefore limits their utility. For example, GBM cells have been shown to secrete VEGF following radiation [83]; in a dish that does not contain endothelial cells, any radio-protective aspect of the subsequent tumor-endothelial interaction is not accounted for, likely contributing to false positive for new drugs.

Microglia, the brain’s resident macrophage cells, participate in a variety of processes related to tumor growth. It has been known for almost 100 years that microglia infiltrate throughout GBM [84]. Recent work has highlighted that these cells actively promote glioma invasion through a variety of mechanisms, including activation of matrix metalloproteases (MMPs) [85] and secretion of transforming growth factor-β [86,87] and EGF secretion [88] and through synergistic GBM-derived colony-stimulating factor-1 (CSF-1). Other groups have also shown that activation of microglia by tumor cells leads to secretion of factors that promote GBM progression [89]. Indeed, blockade of CSF-1R on macrophages is sufficient to reduce tumor growth and increase survival in mouse models [90]. These studies show the great influence of tumor associated microglia on tumor progression and invasion. The fact that microglia—like all the other mentioned non-tumor cells—are excluded from current culture systems means that in vitro experiments fail to account for an important set of variables that impinge upon GBM biology.

#### 2.3.5. Dynamic Surroundings

All of the factors limiting the efficacy of current culture systems share a common element: they are dynamic. Brain circuits are constantly changing in response to the demands of the environment and during the processes of cognition. The release of secreted factors like neurotransmitters and neurotrophic factors into the tumor microenvironment will therefore be constantly changing. Tumor monocultures, even with the best-intentioned additives, simply cannot match the second by second variation that occurs in human brain tissue. Take, for example, the way in which growth factors are added to tissue culture experiments. At the time of seeding cells, high levels of EGF and bFGF are usually added to the media, a sort of “bolus injection” of growth factor, which is then slowly utilized or degraded. In contrast, a developing tumor in the cortex will be constantly exposed to different growth factors and secreted molecules. The dynamic nature of the brain microenvironment cries out for novel culture techniques and devices to increase the quality and relevance of GBM research.

## 3. Novel Strategies for GBM Cell Culture

The studies highlighted thus far provide a multitude of reasons to doubt the efficacy of current culture systems. However, they also indicate specific variables in the design of in vitro practices that could be manipulated and optimized to develop better mechanisms for investigating radiosensitizing compounds in GBM. While the sheer multitude of variables is intimidating, a variety of groups have responded to these challenges with innovative and creative solutions. Table 1 summarizes the exciting new advances in GBM culture systems, their advantages, and some remaining challenges. 

### 3.1. Suspension-Based Culture Models: Neurospheres and Organotypic Glioma Spheroids

Growing GBM cells in suspension as neurospheres, typically in the presence of EGF, has gained some traction in recent years [91]. This technique has three distinct advantages. First, it allows tumor cells to grow in three dimensions, meaning that, other than the outermost layer of cells, they will be contacting neighbors on all sides, as in actual tumors. Second, evidence suggests that these neurospheres better preserve the gene expression profile and genetic status of the patient tumors from which they were derived [11]. Third, in single-cell / cell limiting dilution assays, cells that have the ability to grow spheres indicates that the GBM cells are maintaining the ability to initiate tumor formation. Therefore, neurospheres can serve as a useful system for analyzing compounds capable of limiting tumor growth.

Neurospheres, however, are still a somewhat limited system, because they fail to account for many of the microenvironmental factors discussed, many of which are known to influence the sphere forming capacity of GBM cells. Further, suspension culture still introduces the variable of selection to the generation of patient derived cultures. The conditions encountered in vitro will not match those of the tumor milieu, and may artificially promote the survival and sphere forming capacity of certain subpopulations. It has been shown, for example, that GSCs from the edge of tumors are often highly proliferative, while GSCs from the necrotic core are more quiescent [102]. It is likely that certain GSCs will outperform others in sphere cultures, biasing any further experiments. Finally, in our own observations, molecular targeted therapies that have failed to exhibit therapeutic efficacy in GBM clinical trials such as erlotinib and the anti-diabetic biguanide phenformin exhibit cytotoxic activity in neurosphere formation assays compared to other in vitro models (Figure 2), suggesting that anti-glioma compounds identified with this system might not translate in the clinic.

In addition to 3D spheroids containing only GBM cells, organotypic spheroids derived from biopsies and comprising multiple cell types offer an alternative strategy for preclinical tumor modeling. Developed initially in 1990, these spheroids are generated directly from tissue taken from patients during biopsies or resections. They maintain several key aspects of the tumor microenvironment, including intact blood vessels, tumor-associated extracellular matrix, and macrophages [92] and have been reported to preserve the genomic characteristics of the original tumours [93]. In light of the previously discussed importance of the tumor microenvironment, and the non-malignant cells within it, these spheroids have advantages as preclinical models for therapy development. For example, organotypic spheroids have been utilized to study the effect of radiotherapy on GBM [103], including the role of hypoxia in radioresistance [104]. Recent work has continued to highlight their potential for GBM drug discovery, with a Dutch group demonstrating that 3D spheroids more accurately predict the efficacy of radiosensitizers than conventional models, and have superior modeling qualities for investigation of combination therapies [94]. 

New techniques for the establishment of organotypic spheroids continue to expand, including the use of ultrasonic aspiration during tumor resection; spheroids derived in this manner maintained expression of several glioma stem cell markers known to drive aggressive recurrence [95].

The main limitations of these organotypic spheroids are the high cost of establishment and maintenance and their lack of suitability for high-throughput screening approaches [105]. Because they rely on rapid utilization of freshly-isolated tumor samples, they are time and cost intensive. These constraints indicate that organotypic spheroids should perhaps be utilized further along the drug discovery process as a critical test to determine whether compounds identified by less biologically relevant, high throughput assays should progress to in vivo evaluation and eventual clinical testing.

### 3.2. Hydrogels

A variety of hydrogels have been fabricated from a variety of ECM-derived polymers trying to mimic the biochemical composition of the GBM microenvironment. These include hyaluronic acid [106,107], chitosan [108], chondroitin sulfate polysaccharides [109], alginates [108] and collagen/gelatin proteins [106,110]. The use of these hydrogels in GBM has been extensively reviewed by Xiao [111], so will not be discussed in this review any further. Matrigel is a natural hydrogel of animal origin that has been extensively used in the GBM field both as a coating-material or as a 3D growth system and will be discussed further in the sections below.

### 3.3. Matrigel-Coating for 2D Growth

One of major changes in cell culture for GBM has been the addition of different coatings for plastic flasks. In theory, these coatings limit the effects of stiff plastic surfaces on tumor cells, as well as allowing them to occupy an environment with interactions on multiple sides. Matrigel is one of the most prevalent coating materials for GBM cells. Derived from the extracellular matrix of mouse sacroma tumors, Matrigel contains a mix of collagen, laminin, and ECM-associated growth factors [112].

One shortcoming of Matrigel coating is the fact that it is derived from sarcoma tumors and predominately comprised of collagen and laminin. As discussed above, the ECM surrounding GBM tumors bears little similarity to the ECM of other tumor types. While better than simply growing brain tumor cells directly on plastic, the composition of Matrigel introduces a variety of variables that may skew the results of in vitro experiments. Nevertheless, as described above, GSC preferentially occupy the perivascular niche. Since laminin is one of the main constituents of blood vessels’ walls, supporting the use of Matrigel-coated plastics for the growth of GSCs.

Another limitation of Matrigel is poor quality control. Because it is extracted from mouse tumors, it is impossible to ensure that each batch contains equal amounts of the various components. This fact introduces a major variable across experiments, as one flask may have more growth factor than another. While imperfect, Matrigel, both for flask coating and as a 3D substrate, remains a key tool in the current in vitro toolkit, because of its ease of use and the fact it better accounts for the tumor microenvironment than plastic flasks alone.

### 3.4. Matrigel Plugs for 3D Growth

Cells growing in 3D laminin and collagen rich gel were more resistant to radiation and recapitulated the DNA-damage response of xenograft tumors. Mechanistic investigation revealed that a large portion of this enhanced radioresistance relied on changes in chromatin structure [15]. As chromatin structure has been indicated as a player in how GSCs overcome therapy [113], this culture system has great potential as a tool for developing clinically effective radiosensitizers in carcinoma models. In glioblastoma, GSCs grown in Matrigel plugs display an elongated fibroblastic morphology and fail to recapitulate key GBM characteristics such as invasion [18] (and personal communication Dr. Joanna Birch). Extensive analysis of the morphology and radiation responses of these cells in our laboratory has indicated this tool is not suitable for GBM models.

### 3.5. 3D Scaffolds

Another strategy for improving the efficacy of in vitro models of GBM is the use of three-dimensional (3D) scaffolds. A recent report showed that Matrigel-coated 3D polystyrene scaffolds (3D-Alvetex; Figure 2) provided an excellent mechanism for predicting the clinical efficacy of anti-GBM modalities; screens using these 3D cultures correcting identified temozolomide and bevacuzimab as capable of limiting GBM growth, while indicating no efficacy of erlotinib, both alone and combined with radiation [18]. GSCs grown in these 3D scaffolds exhibited morphologies present in human tumours such as microtubes [114] and produced invasion patterns more similar to GBM tumors in brain slices (Joanna Birch, personal communication). This model has two limitations. First, substrate stiffness which can infuence phenotype, proliferation and differentiation in some cell types [115]. These effects have been shown to be overriden by increasing seeding density in a human mesenchymal stem cell 3D model using poly-acrylamide gels [116], suggesting that intercellular distance regulates cell morphology and fate. Second, cell retrieval from polystyrene scaffolds is never 100% effective, and cells retrieved may artificially select subpopulations with more invasive capacity or those that are more loosely attached to the matrix, making this system inadequate for cell maintenance or for experiments requiring cell suspensions such as fluorescence activated cell sorting (FACS). Another group recently demonstrated that a 3D collagen scaffold enables growth of GBM cells that better recapitulate the gene expression of GSCs, as well as more faithfully reflect how tumor cells respond to alkylating chemotherapy [117]. Furthermore, a group from the University of Alabama is currently developing a microtumor 3D culture system for patient derveid samples, using a humanized matrigel alternative. Their research suggests that this model more effectively predicts kinase activity and signaling pathway status than 2D models [118]. As with matrigel, the ECM engulfing GBM tumors contain little collagen, rendering this system as a suboptimal representation of GBM tumor microenvironment. More experiments are needed to further validate if these models can predict translational activity in GBM.

While these scaffolds are still purified monocultures of GBM cells, they appear to have a great advantage over 2D flask cultures in terms of generating cultures that reflect drug sensitivities of actual patient tumors. They also have a great advantage of some of the other new systems discussed here, in that 3D scaffolds appear to be easily and relatively cheaply applied to high-throughput systems, including 96 well plates (e.g., Alvetex). These scaffolds allow researchers to continue to use the same basic culture techniques, while improving the efficacy and accuracy of screens. This ease of use may mean that 3D cultures represent an ideal early stage or massive compound library screening tool, results of which are then confirmed in a more complex model.

In addition to 3D scaffolds made of collagen and polysterene, a growing body of work on biomaterials may provide new systems for culturing GBM cells (reviewed [119]). As discussed above, a major limitation of current culture systems is that they do not create a surface similar to the actual brain ECM. Matrigel, for example, is largely made up of laminin and collagen, two proteins in low concentration in the brain ECM. In response to this discrepancy, new biomaterial scaffolds are under development. Beginning in 2011, the Kumar group developed a 3D scaffold composed on hyaluronic acid (a major component of the brain and tumor ECM) and hydrogel; this scaffold produced invasion patterns more similar to GBM tumors in brain slices than on collagen based scaffolds [120]. More recent work has established that hyaluronic acid rich hydrogel matrices allow for culturing of patient-derived GBM cells from human patients. Cells growing on this matrix exhibited morphology and drug sensitivity more similar to actual tumor samples than cells growing in 2D [109,121]. Recently, these hyaluronic acid scaffolds were engineered to include several key matrix metal proteases from the brain ECM. This system also generated 3D cultures of GBM with phenotypes more reflective of actual tumors [122]. Other groups have generated matrices using hyaluronic acid and chitosan (another key brain ECM component); these matrices more effectively maintain tumor growth patterns and expression of key GSC markers than 2D cultures [108]. These scaffolds have enabled researchers to more fully delineate the effect of the brain ECM on GBM behavior, including invasion [123,124], growth [125], and sensitivity to novel therapies [126,127].

### 3.6. Microfluidic Systems

In order to address the problem of growing cells in static media, researchers have explored the use of microfluidic systems, in which liquid media can be continually passed through growing cells. One exciting example of the use of fluidic chambers in GBM is the generation of glioma cultures for drug discovery came from a recent report, in which GBM primary cells were cultured in a network of hydrogel tubes with circulating media [96]. Cells were pumped into these tubes in a solution containing several brain specific ECM components, including hyalurlonic acid. As the tumor cells grew, they formed small spheres along the edges of the tubes; with longer incubation, they generated continuous strands of cells, marked by high expression of GSC markers. Another novel microfluidic-based system is currently in development at the University of Houston. Termed the “brain cancer chip”, it enables 3D culture of freshly dissociated tumor cells from surgical resection or biopsy specimens. Improved chip design ensures that drugs do not leach between wells, enabling effective drug screening applications [128].

The advantage of these systems appears to be based on the ability to expose GBM cells to CNS-specific ECM molecules and generate high numbers of GSCs. Further, the ability to control the microenvironment surrounding the developing cells via alterations in the composition of the feeding media reservoir could be leveraged to create more dynamic and realistic exposures to key growth factors and other variables, such as acidity. The limitations of this system are two fold. First and foremost, it remains a monoculture, meaning that it fails to account for non-tumor cell contributions to GBM growth. Secondly, it is unclear how this system can be applied to assays and drug screens. The authors themselves seem to suggest that the best use of this system would be to generate high yields of representative GSCs that will then be used for some other drug screening process.

### 3.7. Brain Slices

Another innovative solution for assaying GBM behavior in vitro involves borrowing a technique from the field of neuroscience. Coronal organotypic slices have been a mainstay of electrophysiology studies for years [129]. Briefly, the process involves sacrificing mice and then removing their brains. After dissecting away the olfactory bulb and the cerebellum, brains are sliced on the coronal plane, usual to a width of 200 μM. They can then be kept alive in media for up to several weeks. Groups will then implant GBM cells via nano-injector. This system enables live cell imaging, and has already been used to investigate tumor cell invasion processes [97] and the role of microglia in tumor growth [98]. Another group has used a similar system to show that GBM cells engraft more effectively and grow more quickly in different areas of the brain, indicating microenvironmental forces are key factors governing tumor growth [99].

The key advantage of this method is that it enables experimenters to test hypotheses on tumors that are surrounded by the same populations of cells they would encounter in the human brain. As the mentioned study showed, regional differences in the microenvironment are maintained in coronal slices. Another key feature of these slices, also demonstrated by the referenced studies, is they maintain functional microglial populations, allowing for investigation of the role of these resident macrophages in tumor development [97,98]. There are, however, still limitations to this system, most notably the disruption of blood-flow related factors and the fact that the brain slice and tumor engraftment are bathed in nutrient and oxygen rich media. Despite these issues, organotypic cultures appear to be an excellent tool for investigating mechanisms of microenvironmenal interactions as part of the drug development process.

### 3.8. Mini-Brains

An exciting and emerging platform for modeling glioblastoma tumor in vitro utilizes cerebral organoids—so called “Mini-Brains”. The process of generating a cerebral organoid begins with pluripotent stem cells, which are then slowly grown and differentiated using sequential changes in media and additives. In the final stages, organoids are grown in rotating bioreactors. These mini-brains contain morphologically accurate and functionally mature populations of all brain cell types; they also recapitulate regional differences in the brain [101] (reviewed [130,131]). These brain organoids allow for the investigation of tissue dynamics in real time, including microscope imaging in live cells. Since their creation in 2013, mini-brain modeling has continued to improve, including the generation of microfluidic chip platforms to create perfusion within the organoid-containing dish [132].

This technology has been used to investigate a variety of neurological deficits and is beginning to be applied to GBM. Ogawa and colleagues recently developed two distinct mechanisms for utilizing mini-brains to model GBM development—genetically engineered tumors and orthotopic implantation of patient derived cells [100]. In the first instance, they utilized a duel construct electroporation strategy, in which one construct relied on CRISPR-Cas9 technology altered the fourth exon of TP53 to generate a truncated and non-functional p53 protein. The second construct introduced an oncogenic HRas virus. Co-electroporation of these constructs into the cortical region of mature cerebral organoids led to the development of tumors that recapitulated many of the key features of GBM, including invasive potential and the presence of GSCs. Their second strategy relied on microinjection of patient-derived cells. Two lines were implanted, one which forms tumors in mouse models and one that fails to engraft in mouse cortical tissue; these features were maintained when cells were injected into cerebral organoids. Other groups are currently working to develop similar GBM models, including Howard Fine, who was just awarded NIH funding for the expressed purpose of utilizing cerebral organoids for precision medicine for GBM.

Clearly, mini-brains represent an exciting development in modeling brain cancer. They are not, however, without their remaining challenging. Most pressing in our minds, they lack both endothelial cells and microglia. As discussed, endothelial cells and the blood vessels they form are critical players in GBM progression and therapy resistance, as well as provide a key GSC niche. Experiments in cerebral organoids therefore still fail to account for the role of vasculature and the communication between tumors and endothelial cells. In addition, microglia help regulate tumor growth and invasion. The lack of these cells, as well as any peripheral immune cells, in mini-brains may limit their utility for examining certain therapies, especially immunotherapies. It has been suggested that microglia, derived from pluripotent stem cells separately differentiated, could be added to the mini-brain production process [133], thereby bypassing this issue. Further research will determine how to overcome these challenges. The fact that cerebral organoids can recapitulate the neuronal and astrocytic portions of cortical tissue means they have great potential as in vitro drug-development and drug screening tools for brain cancer.

### 3.9. Tumor Organoids

Building on this exciting work in growing cerebral organoids, other groups have utilized similar techniques in an attempt to develop tumor organoids. Briefly, initial tumor cells (patient derived, patient derived metastases, or genetically engineered) are seeded on small Matrigel pellets. After a few days, they are transferred to a rotation bioreactor, similar to those used to generate cerebral organoids [102]. This group found that tumor organoids maintain oxygen gradients across the tumor, as well as produce a tumor complete with different subpopulations and neo-GSC satellite areas. Radiosensitivity assays also demonstrated that these cultures faithfully mirror the differences in different tumor subpopulations, with differentiated GBM cells dying in response to radiation while neighboring GSCs were resistant. These aspects of the model address several of the issues plaguing simpler in vitro models, including reliably creating a heterogeneous tumor, complete with regional differences in microenvironment and subpopulations differences. While these organoids are still a mono-culture system, and therefore do not account for the influence of non-tumor cells, they have great potential for drug screening assays, especially radiosensitizers.

## 4. Conclusions, Caveats, and Future Directions

The process of in vitro culturing of GBM cells is fraught with challenges. However, as this review shows, innovative and creative solutions are under development to overcome these difficulties. Through the use of new technologies in bioengineering and nano-scale printing, labs are developing new models that better recapitulate the complexity of GBM tumors and their microenvironment. Through continued collaborations across disciplines—from biomaterials experts, to tumor biologists, neuroscientists, and radiation oncologists—these technologies may well unearth new options for GBM patients in desperate need of better therapies.

There remain several potential difficulties with these technologies. First and foremost, none of the discussed culture systems enables researchers to determine how candidate compounds will interact with the blood-brain-barrier. Poor penetrance has been a key factor in the downfall of many promising anti-glioma agents [134,135]. If these new in vitro protocols are to be applied to compound screening and drug development, they will still require groups to investigate BBB permeability after identification. Relatedly, the lack of functional blood vessels means these systems still have a blind spot in terms of how GBM tumors interact with endothelial cells. A large amount of evidence indicating that interactions between these cells and GBM tumors, especially regarding neo-vascularization and GSC niche, are critical to tumor progression and resistance to radiotherapy.

Another key challenge in optimizing preclinical models of GBM is ensuring relevance to clinical treatments. Since brain tumor therapy relies on the combination of radiation and drug treatment, assay design for in vitro drug investigations must faithfully account for the timing and sequence of therapy in actual patients. This point has been persuasively argued by Stone and colleagues [136]. Considering applicability to clinical use from the beginning, coupled with these exciting new models, will ensure that drug discovery and development for GBM is built on a solid foundation.

None of these remaining challenges, however, should dissuade the brain cancer research community from forging ahead in utilizing and improving these new systems. As mentioned, all of them represent substantial improvements over traditional flask culturing. Rather, we hope this discussion of the persistent challenges facing in vitro will inform discussion about the next steps forward.

We suspect that, like many developments in the cancer researchers toolkit, a combination of different systems will be key. As mentioned, certain systems can be more readily adopted by many groups, as well as scaled up for high-throughput drug screening. We predict that tools like 3D scaffolds may be applied to large compound library screens, after which promising candidates are screened using a more complex culture system, such as cerebral organoids. In contrast, researchers investigating more mechanistic insights may find other tools most useful, such as organotypic brain slices for live cell imaging investigating GBM’s marked invasion capacity. These novel culture systems have and will continue to advance our ability to better understand this immensely complicated tumor.

Looking to the future, the applications of the burgeoning field of 3D culture are vast. One exciting prospect of these novel culture systems is their potential use in personalized medicine. By using patient cells collected during primary tumor resection as the basis for generation of tumor organoids or for implantation into cerebral organoids, a multitude of compounds could be screened for each individual patient. Based on existing studies, there is a high probability that these implants or organoids will maintain the intratumoral diversity of patient tumors, thereby creating an opportunity to screen drugs and personalize treatment for each person.

In summary, current GBM culturing systems create vastly over-simplified tumor models. This imperfect models fail to properly predict clinical efficacy of new compounds, leading to wasteful and ineffective trials. Several factors have been proposed to underlie the mismatch between efficacy in the flaks and in actual patients, including artificial selection pressures, the lack of non-tumor cells in culture, and unnatural extracellular matrices and nutrient availability. In response, a variety of new and innovative culture systems are under-development to provide a new system for testing potential anti-glioma therapies. From 3D scaffolds to organoid generation, novel models better recapitulate tumor dynamics and microenvironmental interactions, enabling more accurate understanding of GBM biology. The field of neuro-oncology seems poised to experience a revolution in in vitro drug screening and development, building on these exciting developments to hopefully provide new therapeutic strategies to patients in need. 

## Figures and Tables

**Figure 1 cancers-11-00044-f001:**
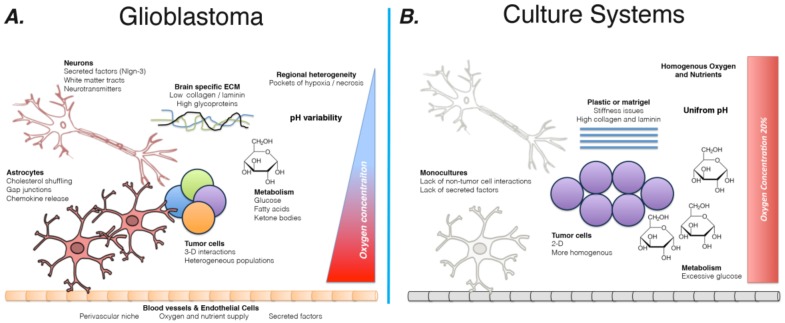
Schematic comparison of tumor microenvironment and current culture systems. (**A**) The brain microenvironment provides a multitude of non-tumor cell types, including neurons, astrocytes, microglia, and endothelial cells. These populations are present in the tumor environment and have been shown to influence the progression of GBM tumors, as well as abet their ability to resist current standard of care therapies, including radiation. In addition, tumor cells in the brain grow in a complex 3D structure entangled with a brain-specific extra-cellular matrix (ECM). They also exhibit a marked amount of intratumoral heterogeneity, with a variety of subpopulations within each tumor. Finally, the tumor microenvironment varies widely across different anatomic locations, with differences in oxygen and nutrient availability; this range of milieus contributes to the intratumoral heterogeneity of GBM. (**B**) In contrast, current culture systems are over-simplified and introduce variables not encountered in the brain. Oxygen concentrations are held constant at 20% by most incubators, while media contains an elevated amount of glucose. Plastic flasks introduce an unnaturally stiff surface, which GBM cells are known to react to. Further, coating flask with matrigel, while better than culturing directly on plastic, exposes tumor cells to elevated levels of collagen and laminin. Mono-cultures mean that in vitro experiment fail to account for non-tumor cells contributions to the microenvironment.

**Figure 2 cancers-11-00044-f002:**
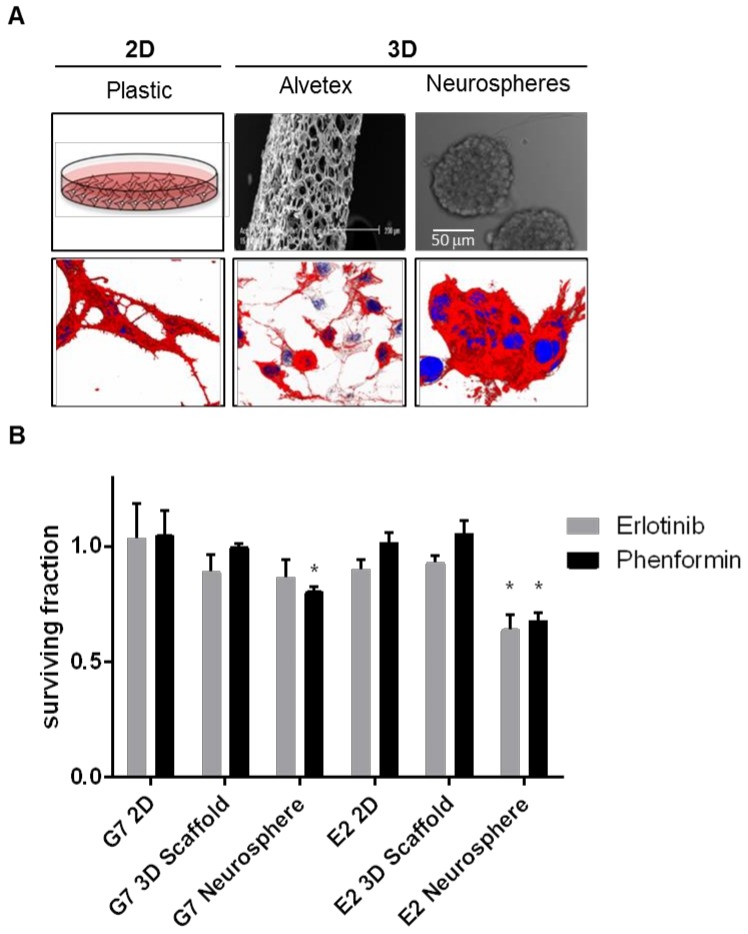
Comparison of erlotinib and phenformin activity on three different cell culture systems. (**A**) Schematic representation (upper panels) and 3D reconstructions of immunofluorescent images (lower panels) of cells stained for F-actin (red) and nuclei (DAPI, blue) of three cell culture systems: 2D growth on plastic plates (left images, 63× magnification); 3D scaffold growth using 3D-Alvetex system (middle images 63× magnification); and 3D neurospheres (right images, 63× magnification, zoom 2×). 2D plastic and 3D scaffolds were previously coated with diluted Matrigel as described in section Matrigel-coating for 2D growth. (**B**) Bar charts representing surviving fraction of cells relative to control grown on different culture systems. G7 and E2 patient-derived GBM cell lines treated with vehicle (DMSO, control), phenformin (10 µM), or erlotinib (1 µM). Drug-treated cells were normalised to vehicle (mean ± SD, of three independent experiments). * *p* < 0.05, relative to DMSO control of their respective culture system.

**Table 1 cancers-11-00044-t001:** Summary of new in vitro GBM modeling systems.

Novel System	Summary	Pros	Cons	Key References
Matrigel Plugs	Tumor cells are grow embedded in a 3D matrigel	Allow cells to grow in 3D Better approximation of the stiffness of the brain ECM ECM-derived growth factors better mirror how GFs are available to the tumor	Mono-culture Sarcoma ECM is not equivalent to the brain ECM (elevated collagen and laminin)	[15]
Neurospheres	Cells are grown in suspension, often in neural stem cell promoting media	Allow cells to grow in 3D Ensure that cells examined are tumorigenic, likely the key target for anti-glioma therapy	Mono-culture	[91]
Organotypic glioma spheroids	Cells from patient resections, grown in 3D cultures	Multiple cells types (including macrophages and endothelial cells) Maintenance of genetic hierarchy within tumor cells	Low throughput ££	[92,93,94,95]
3D Scaffolds	Cells grow in a matrix, allowing for 3D cell interactions	Allow cells to grow in 3D Easily scaled to high-throughput drug screening Can include brain-ECM specific components Better recapitulation of gene expression and invasion	Mono-culture Oxygen and nutrient availability is consistent across the entire population **££**	[18]
Microfluidic Systems	Cells are grown in hydrogel tubes, which are filled with circulating media	Time-dependent exposure to nutrients/more dynamic micronvironment Allow for the generation of many GSCs from a small initial sample	Mono-culture **££**	[96]
Brain Slices	Tumor cells are implanted into mouse brains	Expose tumor to all the neighboring non-tumor cells Great for examining invasion in the presence of white matter and secreted factors	Implantation is always problematic Mouse host with human cells Lack of circulation	[97,98,99]
Mini-brains (transduced)	Cerebral organoids are genetically modified to generate tumors	Expose tumor to all the neighboring non-tumor cells Real-time live cell imaging compatible Allow for examination of tumor initiation and early progression	**££** Time requirements Lack of vessel formation/endothelial cells Currently lack microglia Genetic engerieed tumors inherently involve selecting a driving mutation Not clear which cell type is being transduced during electroporation Still rely on matrigel seed	[100]
Mini-brains (implanted)	Cerebral organoids, once established, have patient derived cell injected into the cortex	High potential for personalization and precision medicine Human to human graft	**££** Time requirements Lack of vessel formation/endothelial cells Currently lack microglia Still rely on matrigel seed	[100,101]
Tumor Organoids	Tumor cells are used as the starting point for generation of a “cerebral” organoid	Maintain nutrient and oxygen gradients Maintain regional intratumoral heterogeneity Appear to promote preservation of cellular phenotype	Mono-culture **££** Time requirements Still rely on matrigel seed	[102]

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
