# Peer review of "Glioblastoma’s Next Top Model: Novel Culture Systems for Brain Cancer Radiotherapy Research"

_cancers, 2019, doi:10.3390/cancers11010044_

Round 1
Reviewer 1 Report
This manuscript review describes the limitations of preclinical (high-throughput) models of glioblastoma and highlights several factors that can and cannot be manipulated in current systems. While not an exhaustive discussion, the article does cover many of the most pertinent points in the literature. The article is a little heavy on matrigel-based models with less attention to other hydrogels and fully human matrices. While some aspects of the discussion describe detail regarding conditions, such as glucose ranges, other aspects, such as O2 content, pH, are limited in detail or not described. Perhaps a table or figure could show the idealized model system that could include those factors (and ranges, such as glucose or pH) that can be manipulated in culture. This would provide an excellent resource for investigators interested in optimizing a model system. Some recent relevant publications have not been cited, but could be added to this review in a minor revision, such as the following PMID's: 29937354 (discussing GBM growth influenced by neighboring cells/extracellular vesicles), and articles using advanced models for drug screening and stem cell growth including 30337660, 29946469, 28993067, 29849102.
Minor typographical errors could be corrected and some word choice could be altered (e.g., the use of the word "Critically" so many times in the manuscript).
Nevertheless, this is a well written and important review that only requires minor edits prior to publication.
Author Response
This manuscript review describes the limitations of preclinical (high-throughput) models of glioblastoma and highlights several factors that can and cannot be manipulated in current systems. While not an exhaustive discussion, the article does cover many of the most pertinent points in the literature. The article is a little heavy on matrigel-based models with less attention to other hydrogels and fully human matrices. While some aspects of the discussion describe detail regarding conditions, such as glucose ranges, other aspects, such as O2 content, pH, are limited in detail or not described. Perhaps a table or figure could show the idealized model system that could include those factors (and ranges, such as glucose or pH) that can be manipulated in culture. This would provide an excellent resource for investigators interested in optimizing a model system. Some recent relevant publications have not been cited, but could be added to this review in a minor revision, such as the following PMID's: 29937354 (discussing GBM growth influenced by neighboring cells/extracellular vesicles), and articles using advanced models for drug screening and stem cell growth including 30337660, 29946469, 28993067, 29849102.
Thank you for taking the time to thoroughly assess our manuscript. In order to improve the manuscript, we have endeavored to make all the changes your requested, including the addition of the references listed here. In the case of the designing an optimized model system, we have expanded our existing schematic to include more details, including pH. We are especially appreciative of the articles on other models for drug screening and stem cell growth.
Minor typographical errors could be corrected and some word choice could be altered (e.g., the use of the word "Critically" so many times in the manuscript).
Thank you for pointing out this error in style. We have made several changes to the wording of the essay, including removing most uses of the word critically.
Nevertheless, this is a well written and important review that only requires minor edits prior to publication.
We appreciate these kind words. Thank you for your input, which we believe has greatly enhanced the manuscript.
Reviewer 2 Report
General comments
Clinical outcome of current GBM therapy is still disappointing, with a poor median survival hence high recurrence rate. In vitro studies and further in vivo validation of effective treatment combinations, basic and mechanistic studies, will ultimately contribute to treatment optimization for this group of patients. The ms. is a comprehensive review on a relevant subject in brain cancer research. It provides a detailed description, listing of pros and contras as well as practical laboratory information regarding the various in vitro models used for preclinical brain tumour research. This in particular for evaluation of the effects of radiation alone or combined with other treatment modalities, such as chemotherapy and targeted drugs. The information provided – about tumour heterogeneity, microenvironmental factors, metabolism, nutrients and oxygen- , is very helpful for researchers in oncology, in particular in the field of radiobiology / radiotherapy.
The manuscript is well written. This reviewer has few but significant comments and suggestions, which are listed below.
Specific comments
Introduction
I do not like the analogy between the process of drug development and selecting a Formula 1 driver. The comparison might be funny for a presentation for students, but not for an introduction in a scientific paper. (Note: this has nothing to do with my love for cycling!). Please delete lines 35-47.
Some parts of the text are very detailed, e.g. regarding the composition of culture media. I agree that this can profoundly influence the phenotype of GBM cells, and therewith the efficacy of culturing of GBM cells. Please consider shortening of the text, e.g. in section 2.1.
Line 173: numbering of the section is not correct. Anyway, I do not understand at all the numbering of the different sections.
Figure 2: Please delete the black bars of the controls. Use e.g. mention in the figure legend “relative to controls”.
This reviewers' serious main concern:
I am missing the clinical relevant so-called “organotypic glioma spheroid” model, which was developed by the Neuro Oncology research group from the Bergen University in Norway. This model, also referred to as ‘biopsy spheroids”, has a large number of pros for preclinical radiation – drug studies. It reflects tumour – cellular – heterogeneity, tumour cell proliferation rate and tumour physiology and can be used in vitro and in vivo. Genetics are preserved (De Witt Hamer et al., Oncogene 2008). This model has been used in multiple studies, all available in the literature, testing radiation – drug interactions. The model is still being used in brain tumour research.
To this reviewers opinion, this in vitro model information has to be added to your review. Data should be added to table 1 and the model discussed in the text of your review. Please see PubMed for the publications, using the key words: “organotypic glioma spheroid” and
Discussion:
Suggestions for the references
I would suggest to include some phrases mentioning that, apart from laboratory technical aspects of in vitro culturing models in radiobiology and drug combination studies, it is of utmost importance to use clinical relevant experimental protocols. See e.g. Stone HB et al. (Translational Oncology 9, 2016), in which great methodological concern is expressed regarding assays and endpoints, scheduling and dosing of radiation-drug combinations etc. (Note: I like that paper and show it often to my students!).
Narayan RS et al. (BMC Cancer, 2017) demonstrated the advantage of using the 3D multicellular glioma spheroid model vs. glioma cells growing in 2D monolayer for preclinical testing of a drug (AKT inhibitor) in combination with irradiation.
You might also mention that in vivo models are recently reviewed by Bristow RG et al. Lancet Oncology 2018.
Regarding the BBB, both references are dated. I would suggest to include the article by Van Tellingen O. et al., Drug Resist Updat. 2015.
Author Response
Clinical outcome of current GBM therapy is still disappointing, with a poor median survival hence high recurrence rate. In vitro studies and further in vivo validation of effective treatment combinations, basic and mechanistic studies, will ultimately contribute to treatment optimization for this group of patients. The ms. is a comprehensive review on a relevant subject in brain cancer research. It provides a detailed description, listing of pros and contras as well as practical laboratory information regarding the various in vitro models used for preclinical brain tumour research. This in particular for evaluation of the effects of radiation alone or combined with other treatment modalities, such as chemotherapy and targeted drugs. The information provided – about tumour heterogeneity, microenvironmental factors, metabolism, nutrients and oxygen- , is very helpful for researchers in oncology, in particular in the field of radiobiology / radiotherapy.
The manuscript is well written. This reviewer has few but significant comments and suggestions, which are listed below.
Thank you very much for examining our manuscript and providing these critiques. We have worked to make all the requested changes.
SPECIFIC
Introduction
I do not like the analogy between the process of drug development and selecting a Formula 1 driver. The comparison might be funny for a presentation for students, but not for an introduction in a scientific paper. (Note: this has nothing to do with my love for cycling!). Please delete lines 35-47.
Thank you for this comment. We apologize if this analogy came across as glib and have removed those lines.
Some parts of the text are very detailed, e.g. regarding the composition of culture media. I agree that this can profoundly influence the phenotype of GBM cells, and therewith the efficacy of culturing of GBM cells. Please consider shortening of the text, e.g. in section 2.1.
In order to make the section crisper, we have shortened it and highlighted only the most relevant portions.
Line 173: numbering of the section is not correct. Anyway, I do not understand at all the numbering of the different sections.
We apologize for the confusing numbering system. We have tried to utilize the hierarchical structure requested by the journal’s editorial standards. In order to reduce confusion, we have adjusted the numbering.
Figure 2: Please delete the black bars of the controls. Use e.g. mention in the figure legend “relative to controls”.
This reviewers' serious main concern:
I am missing the clinical relevant so-called “organotypic glioma spheroid” model, which was developed by the Neuro Oncology research group from the Bergen University in Norway. This model, also referred to as ‘biopsy spheroids”, has a large number of pros for preclinical radiation – drug studies. It reflects tumour – cellular – heterogeneity, tumour cell proliferation rate and tumour physiology and can be used in vitro and in vivo. Genetics are preserved (De Witt Hamer et al., Oncogene 2008). This model has been used in multiple studies, all available in the literature, testing radiation – drug interactions. The model is still being used in brain tumour research.
To this reviewers opinion, this in vitro model information has to be added to your review. Data should be added to table 1 and the model discussed in the text of your review. Please see PubMed for the publications, using the key words: “organotypic glioma spheroid” and
Discussion:
Suggestions for the references
I would suggest to include some phrases mentioning that, apart from laboratory technical aspects of in vitro culturing models in radiobiology and drug combination studies, it is of utmost importance to use clinical relevant experimental protocols. See e.g. Stone HB et al. (Translational Oncology 9, 2016), in which great methodological concern is expressed regarding assays and endpoints, scheduling and dosing of radiation-drug combinations etc. (Note: I like that paper and show it often to my students!).
Narayan RS et al. (BMC Cancer, 2017) demonstrated the advantage of using the 3D multicellular glioma spheroid model vs. glioma cells growing in 2D monolayer for preclinical testing of a drug (AKT inhibitor) in combination with irradiation.
You might also mention that in vivo models are recently reviewed by Bristow RG et al. Lancet Oncology 2018.
Regarding the BBB, both references are dated. I would suggest to include the article by Van Tellingen O. et al., Drug Resist Updat. 2015.
Thank you very much for providing such an extensive set of references for us to add to the manuscript. We have included all of them in the paper, and have added a section on organotypic glioma spheroids as requested.
Round 2
Reviewer 2 Report
To the authors
The authors conveniently responded to most of my questions and critical remarks, took over my literature suggestions, and adapted their ms. accordingly.
This reviewer has however still one main serious point which has not been considered appropriately: the authors should clearly distinguish between multicellular spheroids, composed of one cell type, either established or primary glioma cells, and the so-called “organotypic spheroid model”, spheroids directly prepared from glioma tissue specimens. These spheroids display morphological features similar to those of the original tumor tissue in vivo; in this respect they are different from spheroids obtained from permanent cell lines. The spheroids contain preserved vessels, connective tissue, and macrophages, revealing a close resemblance to the conditions in the original tumor. As mentioned in my comment to the original ms., this organotypic glioma spheroid model, was developed at the Bergen University in Norway and has been described in the next paper:
https://www.ncbi.nlm.nih.gov/pubmed/2406382?dopt=Abstract&holding=npg
This model, also referred to as ‘biopsy spheroids”, has a large number of pros for preclinical radiation – drug studies. It reflects tumour – cellular – heterogeneity, tumour cell proliferation rate and tumour physiology and can be used in vitro and in vivo. Genetics are preserved (the paper of de Witt-Hamer you now included, ref. 92). This model has been used in multiple studies, all available in the literature, testing radiation – drug interactions. Please see PubMed for the publications, using the key words: “organotypic glioma spheroid”.
Thus, the authors should clearly distinguish between the above mentioned model and the – more often – used multicellular spheroid model, which is indeed also used for preclinical testing. In this respect, please notice that ref. 93 does not use the organotypic spheroid model, but the multicellular spheroid model. Indeed, in section 3.1, the description and use for preclinical testing of both experimental models are mixed up, which is confusing.
The organotypic spheroid model information has to be added to your review, for completeness. Data should also be incorporated in table 1 and the model discussed in the text of your review.
Final point: please consider removal of ref. 79. Oncotarget is a predatory journal, which should be stopped. See: https://predatoryjournals.com/journals/
Predatory open-access publishing is an exploitative open access academic publishing business model that involves charging publication fees without providing the editorial and publishing services associated with legitimate journals (open access or not). The idea that they are "predatory" is based on the view that academics are tricked into publishing with them, though some authors may be aware that the journal is poor quality or even fraudulent.
Author Response
We have responded to Reviewer 2’s comments on our revised manuscripts by augmenting and enhancing the discussion of spheroid-based cultures for GBM preclinical modeling in section 3.1 and adding other examples in subsequent sections (all highlighted in bold). We have added ‘organotypic gloma spheroids’ to the title of section 3.1, in order to increase the profile of this type of model. We hope that with these changes, the manuscript will be ready for publication in Cancers.